# Dynamic genetic architecture of yeast response to environmental perturbation shed light on origin of cryptic genetic variation

**Yanjun Zan** [ID]*, **Örjan Carlborg** [ID]

Department of Medical Biochemistry and Microbiology, Uppsala University, Uppsala, Sweden

* yanjunzan@gmail.com

## Abstract

Cryptic genetic variation could arise from, for example, Gene-by-Gene (G-by-G) or Gene-by-Environment (G-by-E) interactions. The underlying molecular mechanisms and how they influence allelic effects and the genetic variance of complex traits is largely unclear. Here, we empirically explored the role of environmentally influenced epistasis on the suppression and release of cryptic variation by reanalysing a dataset of 4,390 haploid yeast segregants phenotyped on 20 different media. The focus was on 130 epistatic loci, each contributing to segregant growth in at least one environment and that together explained most (69–100%) of the narrow sense heritability of growth in the individual environments. We revealed that the epistatic growth network reorganised upon environmental changes to alter the estimated marginal (additive) effects of the individual loci, how multi-locus interactions contributed to individual segregant growth and the level of expressed genetic variance in growth. The estimated additive effects varied most across environments for loci that were highly interactive network hubs in some environments but had few or no interactors in other environments, resulting in changes in total genetic variance across environments. This environmentally dependent epistasis was thus an important mechanism for the suppression and release of cryptic variation in this population. Our findings increase the understanding of the complex genetic mechanisms leading to cryptic variation in populations, providing a basis for future studies on the genetic maintenance of trait robustness and development of genetic models for studying and predicting selection responses for quantitative traits in breeding and evolution.

## Author summary

Many biological traits are polygenic, with complex interplay between underlying genes and the surrounding environment. As a result, individuals with the same allele might have distinctive phenotypes due to differences in the polygenic background and/or the environment. Such differences often create additional genetic variation that is highly relevant to quantitative and evolutionary genetics by limiting our ability to accurately predict the phenotypes in medical or agricultural applications and providing opportunities for long

**Data Availability Statement:** All data used in this study are publicly available in earlier published articles (Bloom et al; https://www.nature.com/articles/ncomms9712; Forsberg et al; https://www.

nature.com/articles/ng.3800. All scripts for analysis and intermediate numerical data for generating figures are available on GitHub: https://github.com/yanjunzan/Yeast783-GGE

**Funding:** This work is supported by the Swedish Research Council (Grant 2017-03726;https://www.vr.se/) to Orjan Carlborg.The funders has no role in the study design, data collection and analysis, decision to publish, or preparation of the manuscript.

**Competing interests:** The authors have declared that no competing interests exist.

term evolution. Previously, yeast growth regulating genes were found to be organised in large interacting networks. Here, we found that these networks were reorganised upon environmental changes, and that this resulted in altered effect sizes of individual genes, and how the whole network contributed to growth and the level of total genetic variance, providing a basis for future studies on the genetic maintenance of trait robustness and development of genetic models for studying and predicting selection responses for quantitative traits.

## Introduction

Cryptic or hidden genetic variation is a type of genetic variation normally not seen but can, as has been shown experimentally in many species, emerge from polymorphisms changing their effects upon genetic (G-by-G interaction; Epistasis) [1–3] or environmental (G-by-E interaction) [4–7] perturbations. The release of such variation could lead to the emergence of extreme phenotypes [8], modifications to the penetrance of common diseases [2], facilitation of responses to artificial selection in crops or domestic animals [3,9] and ultimately affect the capacity of populations to adapt to sudden changes in surrounding environments. It is well known that epistasis and G-by-E are important contributors to cryptic variation of continuous traits [9–11]. More recently empirical results from studies on microorganisms, have convincingly reported G-by-G-by-E interactions as a prevalent and potentially important phenomenon for complex trait variation [12–24]. For example, by experimentally generating combinations of mutant alleles/genes, the environmental dependence of epistatic interactions between genes (G-by-G-by-E) has been shown for binary (presence/absence) phenotypes in *E. coli*, *S. cerevisiae*, *D. melanogaster* and *A. thaliana* [12,13,17–19,25,26]. This environmentally dependent epistasis affected both pairwise [13,20–24] and high-order [14–16] interactions, and resulted in abundant changes in connectivity and output of interaction networks in response to environmental changes for interaction networks built for single nucleotide polymorphisms [17–19] and genes [27–29].

An alternative and complementary approach to engineering combination of alleles experimentally for studying the dynamics in environmentally dependent cryptic variation is to measure how much phenotypic variation is expressed by a segregating population of genetically identical individuals (clones or inbred lines) across multiple environments. The contributing genetic mechanisms can be identified in such populations by mapping the individual locus (G-by-E interactions) and epistatic interactions (G-by-G-by-E interactions) whose effects change across environments. Those variations in effects jointly contribute to the suppression and release of cryptic genetic variation in the population upon environmental perturbations. Understanding the link between the dynamics in the genetic architecture of a complex trait across environments, defined as which loci contribute to the trait in which environments and how their effect sizes/contributed genetic variance change, could provide insights on many aspects of evolutionary and quantitative genetics. For example, as most natural populations/species encounter both subtle and dramatic environmental changes within/across clines/generations, an improved understanding on this phenomenon is central for understanding the genetic architecture, evolution and adaptation of complex traits.

As introduced above, G-G-E interactions have been reported in several species [12,13,17–19,25,26] including *E.coli*, *S. cerevisiae*, *D. melanogaster* and *A. thaliana*, however, how such interactions contribute to rudimentary concepts, such as estimates of variance, allelic effect and others, in quantitative, population and evolutionary genetics has not been explored.

Further exploration on how they influence the phenotypic variance across environments, the estimates of quantitative genetic metrics, including additive/epistatic effects and variances, are central for studies aiming to dissect and predict complex traits. Here, we attempted to provide insights to these questions by reanalysing a panel of 4,390 yeast recombinant offspring (segregants) generated by crossing a laboratory strain (BY) and a vineyard strain (RM). Every segregant was genotyped and phenotyped for growth in 20 media [30]. In an earlier study, Bloom et al [30] reported large differences in the genetic and phenotypic variance of growth among the different growth environments and mapped 939 additive QTL and 330 epistatic QTL when treating growth on these media as independent traits [30]. In these analyses, most of the epistatic QTL were detected via significant pairwise statistical interactions with one or two other loci [30]. Extending this work further, Forsberg et al [31] defined multi-locus networks in the same data by connecting loci with pairwise statistical interactions. Using these networks as a basis, high-order interactions were found to be prevalent and have strong phenotypic effects that could not be predicted from the additive and pairwise epistatic effects. In particular, highly interconnected hub-loci were detected on 11 of the 20 growth media, and they regulated growth by epistatically suppressing or releasing cryptic genetic effects of multiple interactors [31]. Here, we explored how often, and which, connections in the epistatic networks change across environments and how these changes in the networks affected the level of genetic variance in growth displayed by the population (G-by-G-by-E interactions). We explored the same 330 epistatic QTL from Bloom et al [30], but instead of treating growth on each media as independent traits, we here considered growth as a single complex trait measured in different media. These growth measurements on different media then represented expression of the same phenotype in multiple environments where differences in the content of the growth media were considered as various types and strengths of environmental perturbations. Across-environment analyses were performed to explore similarities and differences of allelic effects as well as multi-locus networks across environments to reveal the dynamics in the genetic architecture of segregant growth, leading to the suppression and release of cryptic genetic variation, in response to the wide range of perturbations.

Consistent with previous studies [12–24], extensive changes in the connectivity of epistatic loci were found across environments. Furthermore, our analysis found that these were often associated with changes in how the loci contributed to growth. Alterations to the growth environment often resulted in changes of output from network hubs as the number of active interactions changed, leading to altered patterns of epistatic suppression and release of genetic effects from radial loci. As a consequence, the level of suppressed or released cryptic (hidden) genetic variance changed. G-by-G-by-E interactions were thus an important mechanism by which the population modulated cryptic genetic variation by deactivating epistatic network interactions in some environments and activating them in others. The ability of a population to alter the output from epistatic network interactions to release or suppress genetic variation in response to environmental changes facilitated large phenotypic changes and could result in unforeseeable selection responses. The potential impact of these findings on the genetic maintenance of trait robustness in individuals and populations, release of selectable genetic variation by environmental perturbations, the development of genetic models for quantitative traits as well their implications in areas where this might be of practical importance, such as plant breeding and evolution, are discussed.

## Results

To illustrate the principle of our analysis approach, we provide a schematic example showing a hypothetical total interaction network with four loci (Fig 1A; locus id's A-D). This was created

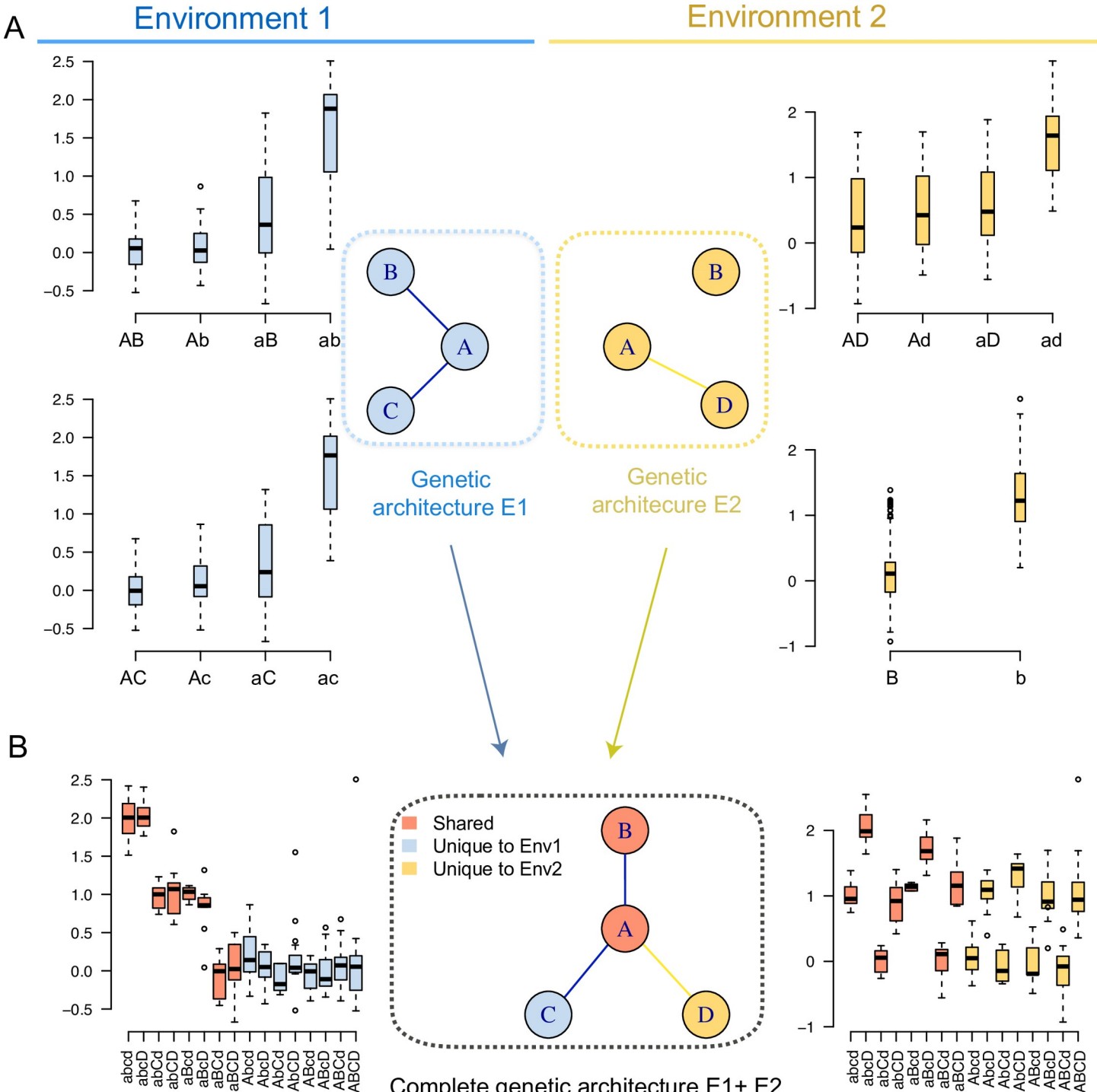

**Fig 1. A schematic illustration of how an epistatic, across environment QTL network is defined and its phenotypic effects in response to environmental perturbations are evaluated.** Epistatic networks are constructed based on results from association analyses modelling two-locus interactions. The circles represent QTL (A-D) involved in significant pairwise interactions contributing to growth (**A**; blue/yellow in environment 1/2, respectively). Interaction networks are first constructed for the environments separately by connecting significantly interacting loci (**A**; blue/yellow lines). The complete set of loci and pairwise interactions are next used to define a complete across environment growth interaction network (**B**). All the within-environment networks are therefore sub-networks. The multi-locus genotype-phenotype maps for the within-environment (**A**) and across environment (**B**) networks are illustrated using box-plots where the growth of the segregants relative to that in a standard environment (y-axis) are given for the evaluated single- or multi-locus genotypes (x-axis). Two networks are evaluated in (**A**) showing how loci contribute to growth in each of the two environments, resembling the results reported in earlier studies of this population [31]. In **B**) the additional analyses performed here to explore the dynamics in the genetic regulation of yeast growth in response to environmental perturbations are illustrated. Here, the complete across-environment interaction network

is used as basis and its multi-locus genotype-phenotype maps explored across all environments. This facilitated both the identification of how the contributions by individual interactions in the network changed across environments (overlaps of loci/interactions across environments), and what the associated changes are in the genetic effects of individual and combination of alleles (differences in the genotype-phenotype maps).

by combining two smaller networks that were independently detected in their own environment (Fig 1A; Environments 1 and 2 in blue/yellow, respectively). On the left and right are the genotype-phenotype (G-P) maps for the pairwise interactions in environment 1 (left; two interactions; Fig 1A), and the G-P maps for one pairwise interaction and a single locus association in environment two (right; Fig 1A). The complete four locus network including all these loci are presented in Fig 1B together with its G-P maps in the two environments (left/right for environments 1/2; Fig 1B). The nodes in this complete network are the mapped loci (QTL) connected by their significant pairwise epistatic interactions (edges). Only loci that have a significant epistatic interaction in at least one of the environments are included. The genotype-phenotype maps (Fig 1A and 1B) illustrate how the estimates of means and variances across the 2/4/16 genotype-classes (defined by one/two/four loci) change in the population of segregants across environments as a result of the activation/deactivation of epistatic interactions in the network. This approach was extended to build a complete yeast growth interaction network (Fig 2A) from all epistatic loci detected in earlier analysis of the 20 environments separately (individually highlighted in S2 Fig) [30,31]. In the following sections, we explore how the activity of the interactions in this complete epistatic network changed across the evaluated environments and how this resulted in changes in classic quantitative genetics measures of the contributions of the involved loci to yeast growth.

## Defining the complete across environment epistatic network of growth loci

The complete across environment growth interaction network included 130 loci, each detected with significant epistatic effect on growth for at least one environment in earlier analyses of this population [31] (Fig 2A, S2 Fig; Materials and Methods details how these networks were constructed; S1 Note). Among them, 69 (53%) were detected in only one environment by their epistatic effects, while the remaining loci were detected with significant epistatic interactions for more than one environment (Fig 2B; S3 Fig). In addition to their epistatic effect, 102 (78%) of the loci were also detected with significant additive effects for at least one of the 20 environments (Fig 2C). It should thus be noted, that some of the 130 loci were earlier only considered as additive loci in one or more of the environments as they only had significant additive effects in them [30]. In total, across all 20 environments, 212 pairwise interactions were detected among the 130 loci. The majority of the interactions (75%, 160) were unique to one environment, while the remaining 25% (52) were detected in at least 2 environments (Fig 2D). When fitting an additive-only model to the 130 loci, one environment at a time, these loci altogether explained between 69–100% of the additive genetic variance for growth in the 20 environments (Fig 2E). This indicates that epistatic interaction and allelic effects are highly dynamic with loci turning on/off their additive/epistatic effects or alter their epistatic connectors across environments.

## The dynamics in the genotype-to-phenotype maps for growth are associated with environmentally induced changes in activity of epistatic network interactions: An example across three environments

We explored the association between changes in the activity of epistatic network interactions and the phenotypic output of additive and epistatic effects of loci, across environments. To

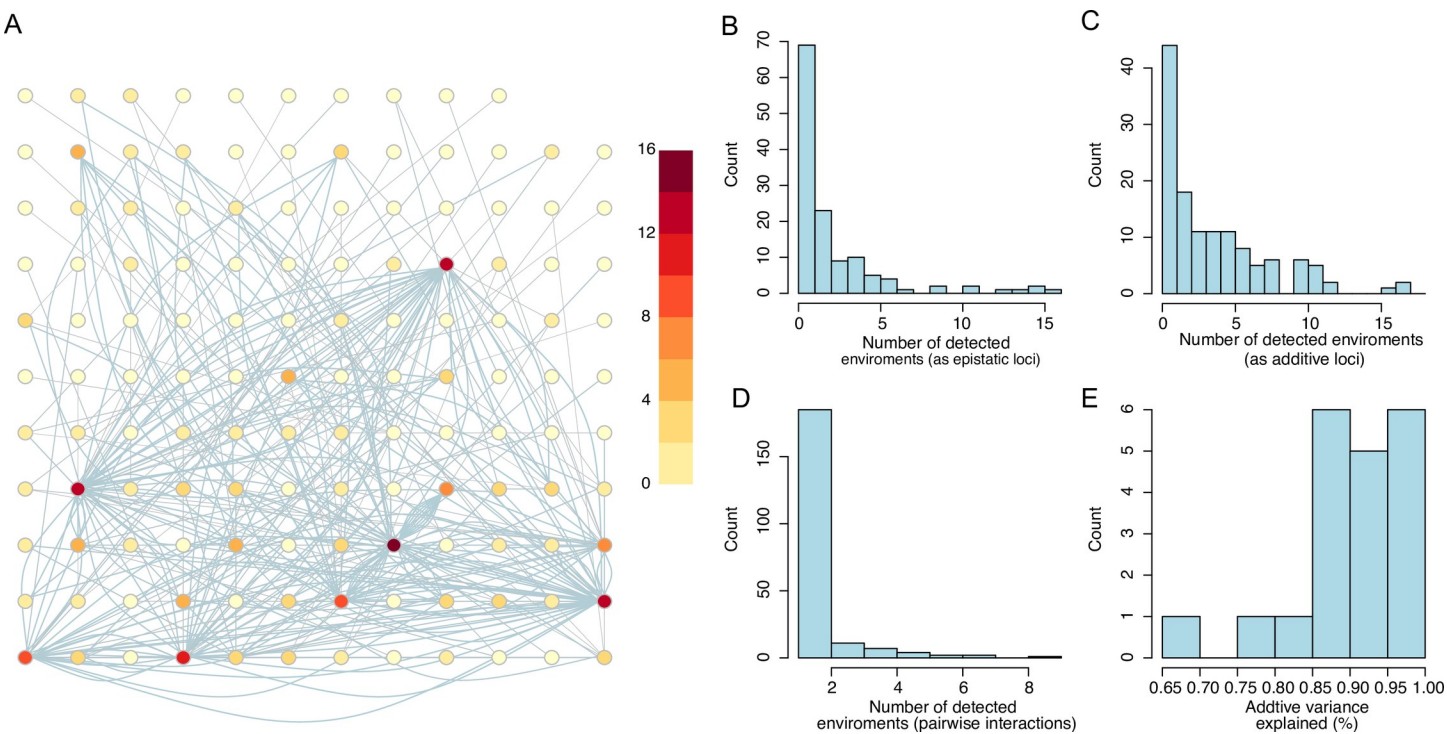

**Fig 2. Summary of the features of the complete interaction network for yeast growth across the 20 studied environments. (A)** The complete across-environment interaction network includes 130 loci contributing epistatically to growth in at least one environment. Each node represents a locus, and its colour shows in how many environments it was involved in significant epistatic interactions. The edges represent significant pairwise interactions between loci, with the number of edges connecting pairs of loci corresponding to the number of times this pair was detected across the 20 environments. **(B/C)** Histograms showing the number of environments in which the 130 loci were involved in at least one significant epistatic interaction/ additive effect. **D)** Histogram showing the number environments in which each of the 212 pairwise interactions were significant. **E)** Histogram showing how much of the additive variance for growth in the 20 environments that was explained by the 130 loci in the network.

achieve this goal, we first present a sub-network built from 3 selected media, where either Indoleacetic acid (IAA) or Formamide had been added, or the carbon source had been changed from Glucose to Raffinose. Segregant growths were, overall, more similar on the media containing IAA and Formamide than in the medium with Raffinose as carbon source (PCA in Fig 3A; pairwise correlations S4 Fig). These media were selected to allow an evaluation of the relationship between degree of similarity of growth environment (measured by the phenotypic correlations in growth) and the resemblance of the underlying genetic architecture (as dissected in the following section). A 28-locus interaction network was defined from the epistatic loci detected for growth on the 3 selected media/environments (as described above and in Materials and Methods). Out of the 28 loci originally detected for their epistatic effect on at least one of the environments, 6 loci have significant additive effects in all 3 environments while another 7 loci have significant additive effects in 2 environments. Similarly, 6 loci have significant epistatic effects in all 3 environments while another 9 loci have significant epistatic effects in 2 environments (Fig 3C). Out of the 42 pairwise interactions detected in the 3 environments, only 1 pairwise interaction is shared across all 3 environments, with another 9 interactions being shared across 2 environments. This changes in activity of the epistatic interactions in the network when the environment changes thus involved (Fig 3E–3G) i) loci that were epistatic in all environments but with changes in the set of loci they interacted with, ii) loci that were epistatic in one environment, but contributed by additive effects only to growth in another or iii) loci that were epistatic in one environment and did not contribute to growth at all in the other.

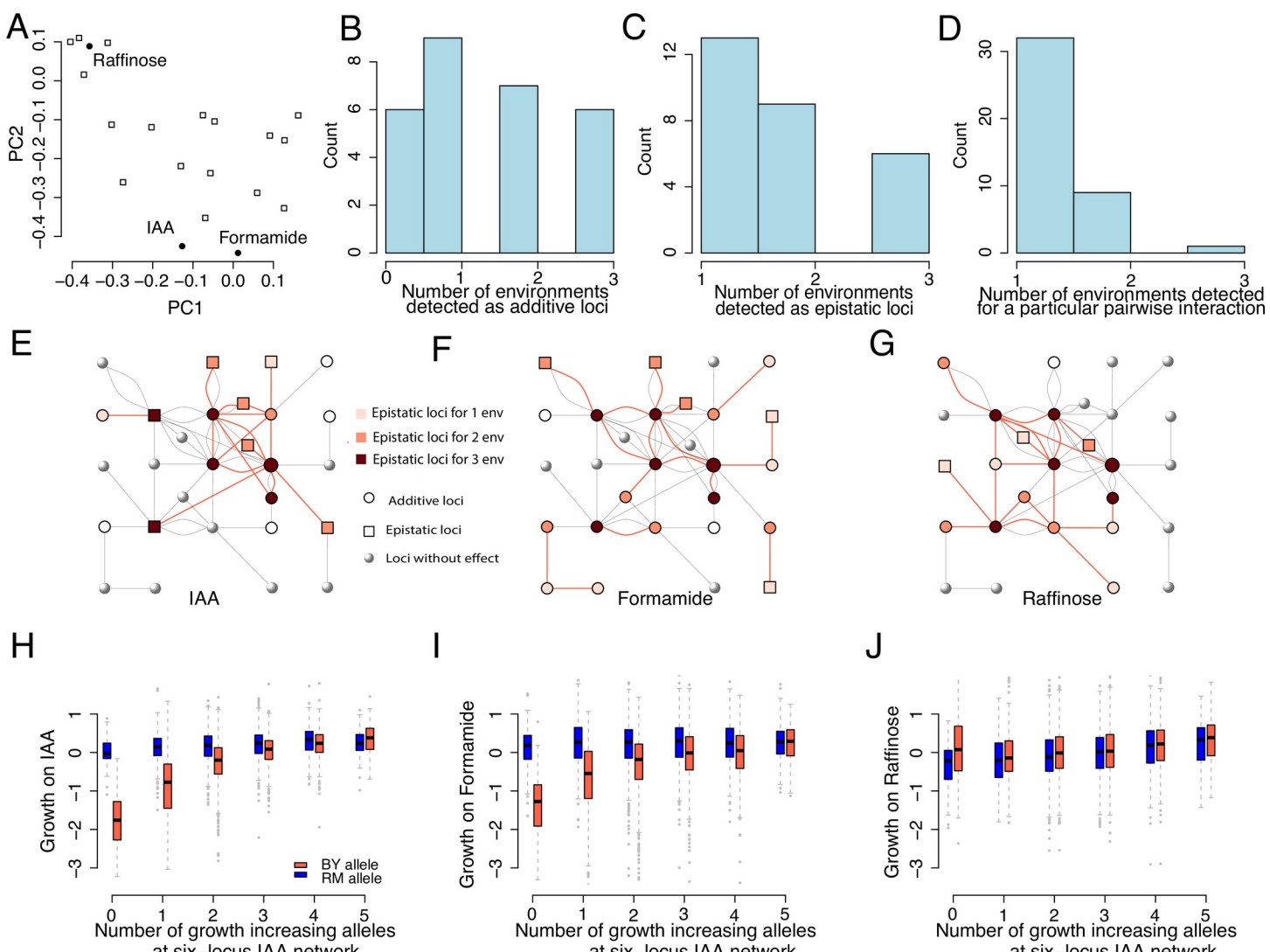

**Fig 3. Illustrations of the relationships between environmental perturbations and contributions by high-order interactions contribute to yeast segregant growth.** A) A two-dimensional PCA plot illustrating the resemblance in growths of yeast segregants across the 20 environments (media). Open squares represent a medium and the filled dots media with added IAA or Formamide or the medium where Raffinose was the carbon source. B/C) Histograms showing the number of environments in which the 28 loci had at least one significant epistatic interaction/additive effect. D) Histogram showing the number environments in which each of the 42 pairwise interactions were significant. E-G) Illustrations of differences and similarities in types of genetic effects (epistatic, additive or none) and activity of the interactions in the epistatic growth network on IAA, Formamide and Raffinose, containing media, respectively. The hub and radial loci in the IAA epistatic network studied earlier [31] are highlighted with red/black arrows. The red lines connects significant pairwise interactions reported in Bloom et al [30]. The grey lines indicate epistatic interaction detected in other media. Panes H-J) Each box plot represents a group of segregants with the same number of growth-decreasing alleles at the five radial QTLs, separated and colored based on the genotype at the hub QTL. The x axis gives the number of growth-decreasing alleles at the radial QTLs and the colour of the box indicates the genotype at the hub QTL (chrVIII: 98,622 bp; tomato/ blue corresponds to BY/ RM alleles).

This is consistent with previous findings [17–19] that environmental changes were associated with extensive changes in interaction networks. To evaluate the connection between the changes in activity of network interactions and the contributions by the loci to growth, a smaller sub-network defined by the locus that was most extensively rewired was explored in more detail (Fig 3E–3G). In total, 13 loci in the complete network contributed epistatically to growth in the medium containing IAA. One of these loci interacted with 7 other loci, defining an eight-locus radial epistatic network for growth. The multi-locus effects of the six strongest

loci in this radial network on growth in IAA-containing medium were explored in detail in earlier analyses of this data [31] (Fig 3E). Here, we found a significant interaction between the 64 multi-locus genotype classes defined by these loci and the three environments (P < 2.2x10$^{-16}$), suggesting a connection between the activity of the interactions in the network and the effects of the loci on growth. The highly connected hub-locus in this network was earlier shown to capacitate the effects on growth from the radial interacting loci [31] (Fig 3H; here capacitate means that the hub-QTL could hide/release the effects of the alleles at interacting loci, resulting in a difference in the narrow sense heritability for alternative genotypes at the hub-QTL as well as difference in the allelic effects of the radial loci between alternative geno-types at hub alleles; Fig 3H–3J). When the segregants were grown on a medium containing Formamide, the output of the epistatic interactions in this six-locus network changed com-pared to that on medium containing IAA. Only 3 of the radial loci remained connected to the central hub. Three loci changed interactivity to now include other loci, one no longer had any effect on growth and interactions between a new locus and the hub was activated (Fig 3E and 3F). Also, the genotype-to-phenotype map for the 6-locus network changed when the segre-gants were grown in media with Formamide. The main difference was a weaker capacitation effect of the hub-locus (Fig 3H and 3I; differences in $h^2$ between the groups of segregants car-rying the alternative alleles at the hub are 0.38/0.13/0.06 on the IAA/ Formamide/Raffinose media, and the difference in allelic effects of the radial loci between alternative genotype at the hub alleles were also weaker). Even larger differences in activity of the interactions, and the genotype-to-phenotype map, were observed when comparing the growth of the segregants on IAA and a medium with Raffinose as carbon source. The hub-locus still contributed additively to growth, and was involved in one new epistatic interaction. It had, however, lost the activity of the interactions with all the interactors contributing to growth on IAA. Two of the original loci no longer contributed to growth. The inactivation of interactions between the hub-locus and its original interactors likely explains its loss of phenotypic effect on growth in this media (Fig 3J). The gradual deactivation of the capacitation (epistatic) effect of the hub-locus from IAA->Formamide->Raffinose was consistent with the phenotypic correlations of segregant growths on these media (PCA plot; Fig 3A; pairwise correlations illustrated in S1 Fig). Together these results illustrate the association between the changes in activity of interactions in the epistatic network and changes in growth effects of the loci involved across environ-ments, i.e. how the connectivity of the hub-locus influences its ability to alter the amount of phenotypic/genetic variance via the radial loci.

### Loci with variable contributions to growth across environments display extensive changes in activity of network interactions: Evaluating effects across all environments

To generalize the findings from the three-environment example above, the contribution by all networks to growth and the associated changes in activity of the epistatic interactions were analysed across all 20 environments. The results are described in detail in the sections below.

**Change in output of high-order gene interaction networks in response to environmental changes is common for highly connected loci.** In total, 13 loci with more than 4 epistatic interactors were detected in at least one of the 20 environments. The networks defined by these 13 hub loci and their interactors included 70% (91 of 130) of the loci in the complete net-work (Fig 2A; S7 Fig). The activity of the interactions of these 13 networks was highly dynamic across the 20 environments, with all interactions to hubs being active in some environments but completely inactive in others (S6A Fig; Fig 4A). Although epistatic network activity changed across environments, it was common that some of the interactors (0–83%; median

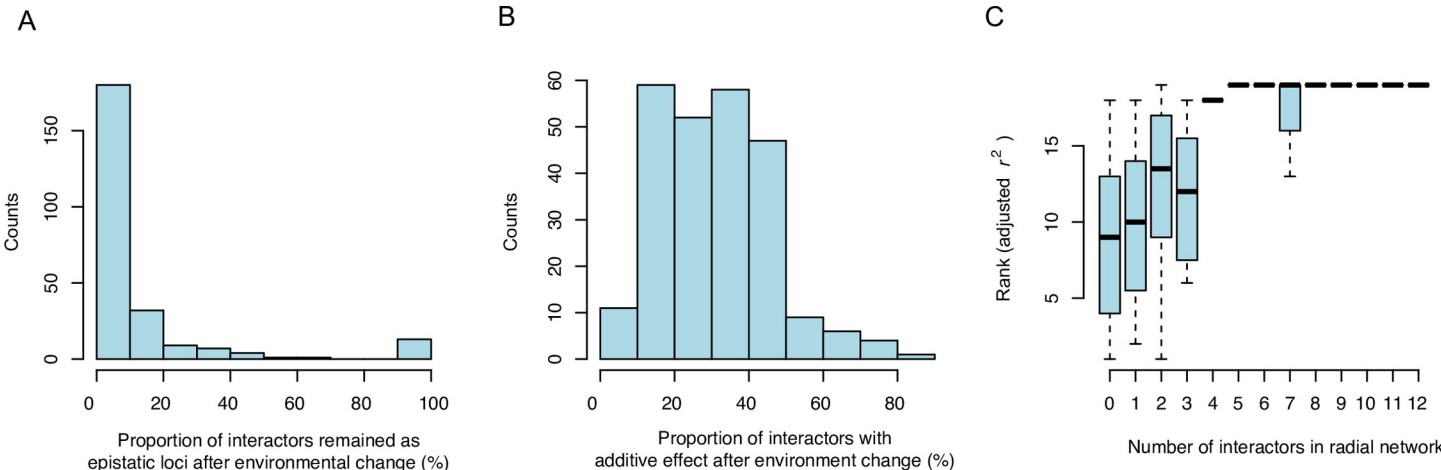

**Fig 4. Illustrations of the dynamics of the interactions in the largest mapped epistatic networks, and their contribution to growth variation, across all tested environments. A)** Histogram illustrating the changes in activity of network interactions in 13 epistatic networks, with more than 4 interactors, across the 20 environments. For each network, a set of interactors was defined to include the radial loci in the most highly activated environment. The percentage of these interactors that were epistatically active with the hub in each of the remaining 19 environments was calculated (x-axis). The overlap of interactors in the 13 networks across the 20 environments was summarized as the counts of environments with similar percentages of shared active interactors (y-axis). **B)** Histogram illustrating the percentage of interactors (defined as in **A** above) that have significant additive effects across the 13 networks and 20 environments. The x-axis shows the percentage of these interactors that have additive effects across the other 19 environments. The y-axis summarizes these percentages across the 13 networks. **C)** In each environment, adjusted $r^2$ values are calculated for all networks and ranks of these model fits were assigned. The association between the connectivity of the epistatic networks (x-axis; number of loci connected to the hub), and their contributions to the variance in growth (y-axis; rank of adjusted $r^2$ values) across the 13 networks and 20 environments, is illustrated as box-plots of these ranks grouped by the number of interactors.

30%) that were inactivated still had detectable additive effects in other environments (Fig 4B; S6B Fig). The power to detect epistatic effects is lower than that for additive effects, making direct comparisons between the number of epistatic interactions and additive effects inappropriate. However, the common observation that epistatic interactions are more frequently inactivated when the environment changes suggests them to be more sensitive to environmental perturbations than additive effects (Fig 5A and 5B; S6A and S6B Fig). In addition, when the interactions between hub and radial loci were active in one environment and not in another, the total contribution by the whole network to the growth variation decreased (Fig 4C). The change of the epistatic networks in response to environment changes was thus associated with their contribution to growth.

**Variations in marginal additive effects across environments associated with the activity of interactions in the epistatic networks.** Significant additive effects on growth in at least one environment were detected for 311 loci in the genome. Although not every locus had significant effects on growth in every environment, these loci did as a group contribute significantly to growth in most environments (Materials and Methods; S1 Table). The effects were generally stronger in one or a few of the environments (S7 Fig). For example, more than half of the loci were unique to one environment and only 9 loci were associated with growth in more than 10 environments. QTL by environment interactions were thus abundant with 98% (307) loci displaying statistically significant QTL by environment interaction after multiple testing corrections (Fig 5; S2 Table; see details in Materials and Methods).

All 11 QTL that were highly active (> 4 epistatic interactors; S2 Table) in at least one environment displayed large variations in the marginal additive effects across the environments (Fig 5A; P-value = 6.1 x 10$^{-8}$; Wilcoxon rank sum test). As the network activity of these loci also changed much between environments, we hypothesised that the additive effects in the different environments were associated with changes in the number of active G-by-G interactions

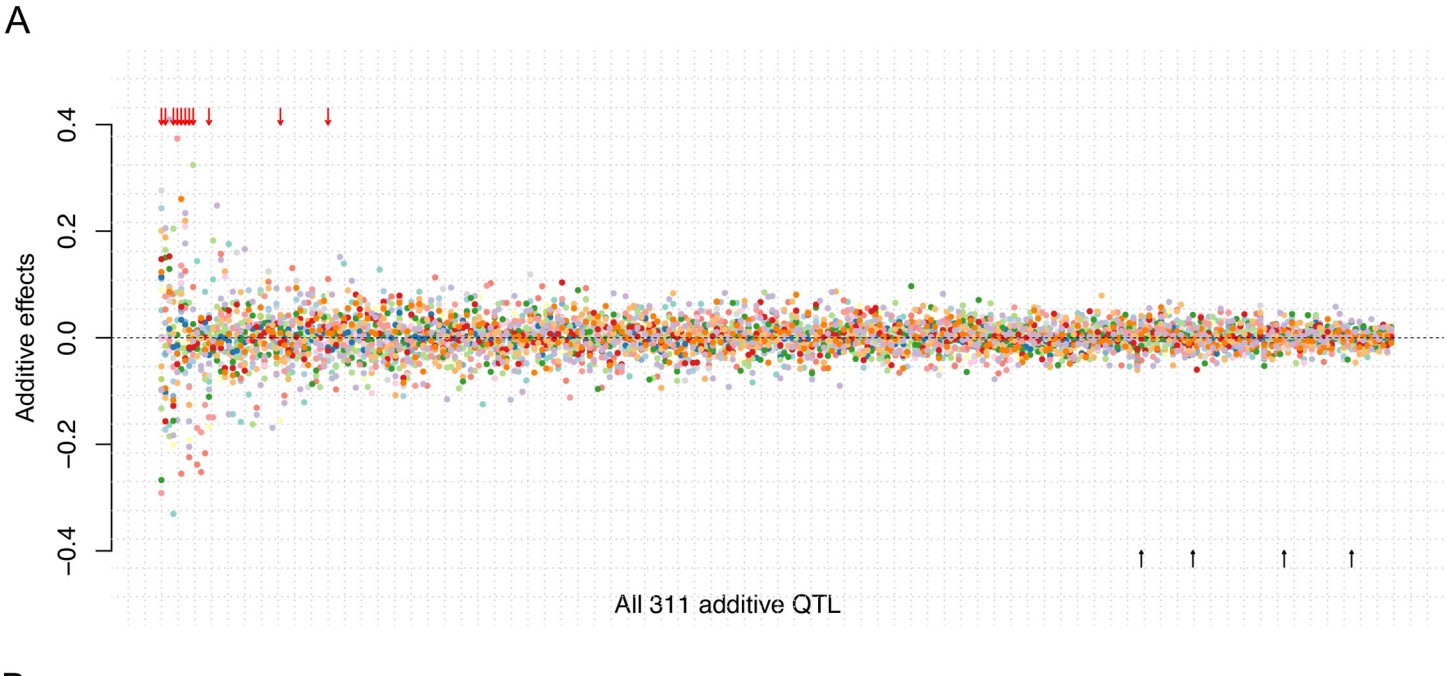

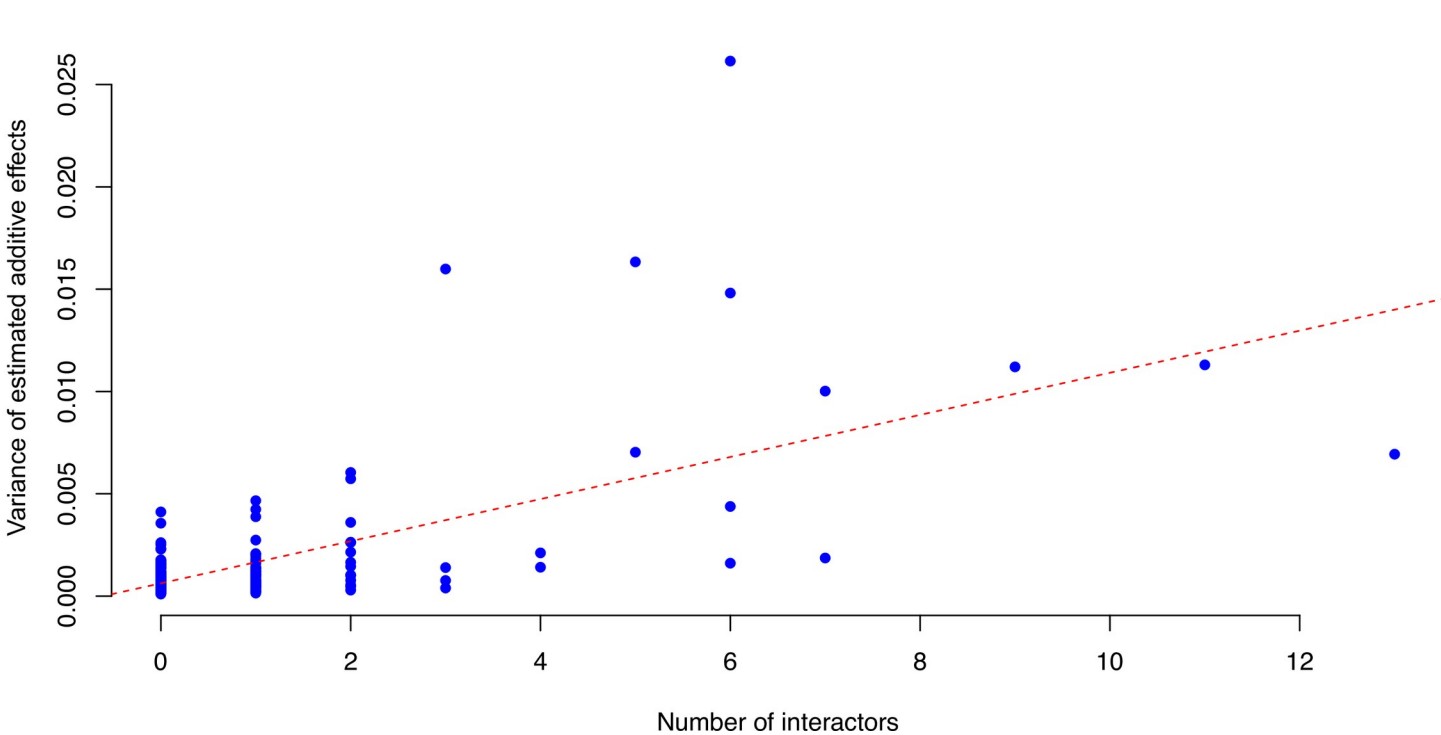

**Fig 5. The variations in additive effects on segregant growth and their associations with network activity. A)** 311 loci had significant additive genetic effects on growth in at least one of the 20 environments. The estimated additive effects (y-axis) for each locus (x-axis) in the 20 studied environments are illustrated using dots in different colours. The loci (represented by the most significant SNPs) are sorted from left to right by the variance of the estimated additive effects across the environments. All additive effects estimates were obtained by fitting all 311 loci jointly in a linear model. The 4 loci without significant genotype by environment interactions are indicated with black arrows below the x-axis. Out of the 13 highly activated loci (interacting with more than 4 other loci), 11 had significant additive effects in at least one environment [31] and they are highlighted with red arrows on the top. **B)** An illustration of the relationship between the maximum number of active epistatic interactions (x-axis) and the variance in their estimated additive effects across the 20 media. Each blue dot represents a locus with a significant additive effect. The red dashed line is the regression line (P value = $7.1 \times 10^{-39}$).

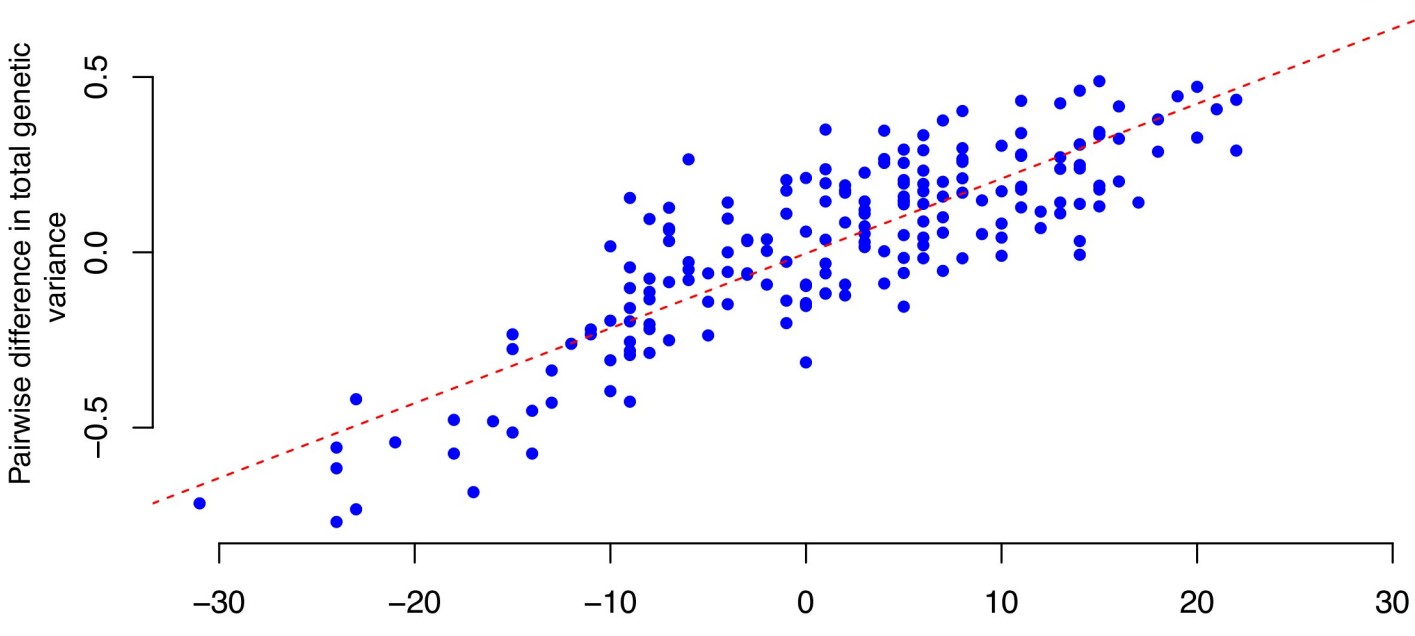

**Fig 6. Illustration of the relationship between the activity of interactions in the epistatic network and amount of total genetic variance explained by the epistatic network across the growth environments.** A significant correlation (P-value = 5.6 x $10^{-31}$ from regression) was detected between the pairwise differences in the number of active epistatic interactions (x-axis) and the total genetic variances ($V_{total1} - V_{total2}$; $V_{total} = H^2$; y-axis) for the loci in the complete interaction network across the 20 environments.

of the locus. To test this, we estimated the correlation between the variations in the number of active epistatic interactors and variance in the additive effects across the 20 environments. This correlation was highly significant (Fig 5B; P value = 7.1x$10^{-39}$; linear regression), suggesting that the variation in additive effects of polymorphisms across the 20 environments, rather than the effect size, was associated with the changes in activity of interactions in the epistatic network.

## Across environment variation in total genetic variance is correlated with changes of the epistatic network

As has been reported earlier in Bloom et al [30], there were considerable variation in both the phenotypic and genetic variance in growth (S8A Fig), indicated by the variation in broad sense ($0.11 < H^2 < 0.88$; median 0.64 [30]) and narrow sense ($0.09 < h^2 < 0.70$; median 0.43 [30]) heritability across environments (S8B Fig). Here, we evaluated whether the changes in the total genetic variance across environments (considered as hidden/cryptic genetic variance) is associated with changes in the underlying epistatic genetic architecture (number of active G-G-E interactions). When studying the changes in the complete 130-locus interaction network across the 20 environments, there are simultaneous forming and collapsing of active sub-networks. Instead of focusing on changes for individual interactions, we instead compared the total number of significantly active epistatic interactions in the environments as a proxy for the total changes in the epistatic genetic architecture. A significant positive correlation (P-value = 5.6 x $10^{-31}$; regression; Fig 6) was detected between the pairwise differences in i) the levels of total genetic variance ($V_{total1} - V_{total2}$; where total genetic variance is estimated as

phenotypic variance times broad sense heritability) and ii) differences in the number of total active epistatic connections ($n_{ep1} - n_{ep2}$) across all pairs of environments. This, together with the results described above, highlights that the dynamics in the underlying genetic architecture, i.e. changes in G-by-G-by-E interactions across environments, is likely to involved in the suppression and release of cryptic genetic variance upon environmental perturbation.

## Discussion

### Approaches to study genotype by environment interactions

One intriguing question in biology is how living organisms cope with changes in living conditions throughout their lifetime. One potential mechanism is Gene x Environment interactions, achieved by gradually modify the effects of common genes, mechanisms or networks that are more or less active across environments. Another form is the recruitment of condition specific mechanisms that are inactive under most of the circumstances. Here, we study this phenomenon by reanalysing a publicly available haploid segregant yeast dataset where growth was measured as colony radius after 48 h growth on various media.

In our study, colony radius was considered as one trait and measurements from the various media were considered realizations of this same traits across multiple environments. A forward genetics screen was first used to identify loci affecting this common growth-measurement across all environments. Subsequently analyses were performed to evaluate where the identified additive/non-additive loci fell on the sliding scale from being common contributors with significant effects in many environments to be loci with more specific in only a few environments. Similar with colony radius, most quantitative traits are likely to be influenced by a variety of mechanisms especially when studied across a range of environments. Hence, rather than studying growth as a complex phenotype resulting from multiple cell biological and physiological processes as in the forward genetic screens, sub-phenotypes that could better reflect the output of the specifically targeted biological process can be assayed separately. Therefore, an alternative reverse genetics approach could be applied to study Gene x Environment by focusing on polymorphisms in these sub-phenotypes targeting specific genes/pathways/networks.

One advantage of the adopted forward genetic approach is that it does not require any prior assumptions about which biological mechanisms are expected to contribute to trait variation, or limit explorations to a set of known genes/pathways/networks. This makes it possible to dissect the genetic basis of quantitative traits using assays that are not specifically targeting the output of individual biological mechanisms and obtain a complete overview of the composite phenotype. In contrast, the reverse genetic approach will provide more detailed insights to specific biological mechanisms beyond what can be achieved in a forward genetics screen. It is, however, difficult to obtain the same general overview of how loci/pathways/networks combine their effects as in a forward genetics screen. Hence, it will likely be a combination of both that gives the direct the future development.

Unfortunately, the reverse genetic approach requires in-depth prior biological assumptions to decompose and measure the composite phenotype as sub-components which are not always available, affordable or even feasible to collect, quantify or pre-define. This hampers the broad application of this approach in quantitative genetics. Another possibility is to use statistics to partition the variance of a complex quantitative phenotype into components aiming at representing common and specific genetic contributions in the studied environments. Separate genome scans could then be performed for these components as independent phenotypes to reveal loci contributing to these. However, such analyses assume that that these statistically inferred components of the phenotypes are biologically independent and further analyses are thus required to obtain a complete view of how they jointly contribute to the composite phenotype.

Despite the large differences in growth environments, high genetic correlations, and considerable overlaps of both additive loci and active epistatic network interactions (S1–S3 Figs) were found. This was common even for growth conditions expected to require distinctive physiological responses, suggesting that underlying signalling and regulatory networks are shared. The forward genetics approach chosen here to analyse the data will not be able to specifically quantify the contributions by defined signalling and regulatory pathways across environments. However, by considering segregant growth as a composite model complex trait resembling those used in quantitative genetic studies in animals, plants and humans, it could provide novel insights to the link between genetic and phenotypic variation in this segregating population. Further methodological development and molecular work is, however, needed to more completely explore and define the roles of individual biological mechanisms in the common and specific genetic contributions to growth in the different environments.

## The effects of genotype-by-environment interactions on yeast growth

The studied dataset did not allow direct estimation of the contribution from environmental (i.e. growth medium) effects to the phenotypes as the available growth measurements were pre-normalised against growth in a control medium [30]. However, estimates of the contributions by direct genetic (G) and genotype-by-environment (G-by-E) effects showed that direct genetic effects only contributed about 1/3 (estimated using ANOVA by fitting the replicate growth measures for the individual segregants across the 20 environments as response variable with the segregant ID and environment as factorial explanatory variables in a linear model) as much to the variance in growth compared to contributions from G-by-E. Consistent with the large contributions from G-by-E, we found large differences in the genetic architecture of growth (defined as associated loci and their genetic effects) across environments. In the following sections we discuss these findings in more detail as well as their implications for the mapping, and understanding the evolution of complex traits.

## High environmental specificity for both additive and epistatic genetic effects

Most of the detected loci, regardless of mapped via their additive effects or epistatic interactions, only contributed to growth in one or a few environments. This in itself indicates extensive G-by-E interactions. However, loci mapped by their additive effects replicated to a greater extent across environments than loci mapped via their epistatic effects (Fig 2, S2 Fig). Epistatic loci were thus highly environmentally specific, suggesting a connection between G-by-G and G-by-E in this population. This is consistent with previous analysis on a similar yeast cross, where cryptic genetic variation is released under rare allelic combinations in specific environmental conditions [19]. The environmentally induced changes in the activity of network interactions influencing the phenotypic output of individual and combinations of loci, altering the level of genetic variance of the population in the environments. Such variations in the allelic effects across environments is thus likely important in many studies of complex traits, for example when aiming to understand the processes allowing individuals and populations respond to changes in the surrounding environments.

## Environmental specificity of high-order interaction effects facilitates buffering of genetic effects in populations

Here, it was shown that the connectivity of the interaction networks, as well as the capacitation effects of the hub-loci, generally were environment specific: the capacitors released the large phenotypic effects of its interactors in some environments, whereas they were often unaffected by them in others. Together with previous findings where yeast growth plasticity was found to

be regulated by environment-specific multi-QTL interaction [17], G-by-G-by-E interactions are thus likely to be a buffering mechanism allowing populations to accumulate cryptic genetic variation in a wide range of environments, for later release in response to environmental perturbations facilitating large and rapid responses.

## Network capacitation influences individual robustness to environmental perturbations

Our results suggest that the interplay between network interactions and environmental factors are important also for individual robustness. Individuals with more non-capacitation hub alleles perform better, on an average (S9 Fig), and tend to show less variability across environments. This might be of relevance to, for example, plant breeding where one of the key challenges is to minimise the impact of genotype by environment interactions on production. Interactions in large gene interaction networks, buffered by environmental factors, might thus be an important driver of observed G-by-E interactions in populations. Targeted breeding for particular alleles at central network hubs might provide routes to genotypes that are either robust performers when challenged by environmental changes or high performers in more defined environments. In addition, G-by-G-by-E might provide multiple routes for populations and individuals to adapt to environmental changes. With the presence of environmentally independent additive effects, it is likely that alleles can be accumulated to intermediate frequencies in populations with small or no fitness costs in many environments. Upon rapid environmental change, the G-by-G-by-E interactions studied here can suppress or release large amounts of selectable genetic variation at a considerably higher rate [32–34], facilitating more rapid and larger selection responses beyond predictions obtained based on the levels of additive variance, or heritability, in the populations.

## Connections to available knowledge

Our study confirms the importance of G-by-G and G-by-E for complex trait variation [9–13,17,31]. For example, many yeast genes are known to be nonessential in one genetic background, but essential in another, with the essential genes often being highly connected hubs in interaction networks [28]. Similar mechanisms have been studied also in bacteria where, for example, evaluations of the effects of 18 randomly selected mutations in *E.coli* in two environments and five genetic backgrounds illustrate that all of them have genetic background dependent effect on phenotypic plasticity [23]. Capacitation is also a well-known mechanisms studied in detail in several species, including the heat shock protein *HSP90* [35,36] in *Arabidopsis thaliana* and *EGFR* in Drosophila melanogaster [37]. A limitation of this study is that it is based on one particular cross, making it difficult to know whether the results are specific to this data or generally applicable. Further work in other crosses, different environments and ultimately natural populations is thus needed to evaluate this. The work, however, provides a strategy for how to design and analyse such new data. Further, given that there is evidence from multiple other species that G by G by E interactions contributes to complex trait variation also present, such work is motivated to more completely dissect also the biological mechanisms underlying such interactions.

## Hypothetical molecular mechanisms underlying the observed effects

A number of the mapped hub-QTL harbour candidate genes with known biological functions, including *GPA1* [38], *HAP1* [39–41], *KRE33* [42], *MKT1* [43] and *IRA2* [44] (S5 Fig). Although they are obvious positional, further work is required to validate the functional candidate genes, in particular if and how they might contribute to the dynamic changes across

environments. Not only because earlier work focussed on marginal genetic effects, rather than network effects, but also as the studies were performed in the environments where the effects were most prominent. Some earlier findings, however, allows us to present hypotheses about ways that they might contribute to the effects discovered here. For example, several hub-QTL were also epistatic hubs for expression QTLs [45] where interactions between *HAP1—KRE33* and between *HAP1—MKT1* contributed to variations in the expression levels of many genes [45]. These studies were, however, performed in a single environment and expression QTL are known to often be environmentally dependent [46,47]. Further studies of transcriptomic and metabolic data across multiple environments would therefore be a possible route to explore whether changes in the connectivity of the genetic networks around these loci across environments results in associated changes also at transcriptomic and metabolic levels.

## Potential implications for modelling of quantitative traits

The studies of this yeast population here, and earlier [31], illustrate that the most highly connected loci in the interaction networks (the hubs) often serve as modulators. They have little, or no, individual effects but rather influence the phenotype by releasing the effects of environmentally specific sets of interacting effector genes. This is an opposite scenario to that assumed in the recently proposed Omnigenetic model for quantitative traits [48], where it is postulated that the highly connected loci in the networks are effectors that are modulated by many other genes. One consequence of this is that results from association studies where loci are detected based on their marginal additive effects need to be interpreted with caution. This is because it might be incorrect to assume that such effects suggest that the locus has a direct (effector) influence on the trait or disease, while it in fact could be entirely a composite effect of contributions by multiple other effector loci. Another potential modelling challenge highlighted here is the potentially large influence from epistasis by environment interactions. We find that they might not only influence the variation in quantitative traits by modulating the effects of individual genes, but also in defining which sets of interactors that are under genetic control by capacitor loci. Further theoretical, and empirical, work is needed to explore the potential implications of these findings for modelling of quantitative trait variation from molecular data in, for example, genome-wide association studies and studies on the basis for, maintenance of and utilization of genetic variation in short- and long-term adaptations to natural and artificial selection. For example, an interesting implication of our findings is that they suggest not only to screen for large effect alleles amongst rare variants [49], but to broaden screens further to facilitate detection of variants that display variable effects across genetic backgrounds (populations) and environments.

## Conclusions

In summary, we show that epistatic networks respond dynamically to environmental perturbations. The dynamics in the network connectivity across environments were connected to changes in allelic effects of individual loci, epistatic effects of multi-locus interactions and the genetic variance contributed by these on the population level. These findings illustrated how G-by-G-by-E interactions influences both individual phenotypes and population level genetic variation. Our results provide novel insights on the fundamental mechanisms contributing to variation in complex traits with practical implications to, in particular, fields where the genetic mechanisms facilitating responses to variations in the environment are central, including evolutionary biology and breeding.

## Methods

### Downloaded data for the BY x RM haploid segregant yeast population

A detailed description of the generation of the 4,390 BY $\times$ RM strains, as well as the genotyping, phenotyping, quality control of genotypes, filtering and normalization of growth measurements is available in Bloom *et al* [30]. All the data analysed here was downloaded from the supplements of that paper. The previously detected additive and epistatic QTLs, as well as their connectivity into within-environment interaction networks, are available in the supplementary information of Forsberg *et al* [31]. The downloaded significant QTL-QTL interaction pairs were originally mapped in a two-locus interaction analysis across all possible combination of genome wide polymorphisms [30]. The within environment interaction networks were constructed by [31], providing input for the definition of a complete across-environment epistatic network.

### Estimating the contributions by genotype and genotype-by-environment interactions to the phenotypic variance

The phenotypic variance was partitioned into contributions from genotype (G), genotype-by-environment (G-by-E) and residual (environmental; E) effects by fitting model (1) to the data:

$$\overline{y}_{ij} = u + id_i + E_j + id_i * E_j + e \tag{1}$$

$\overline{y}_{ij}$ is the mean growth for the replicates of individual *i* in environment *j* (*j* = 1..*n*; *n* is the number of growth conditions); $id_i$ is the individual segregant (genotype) coded as a factor and $E_j$ is a dummy variable representing the growth condition (environment). $id_i*E_j$ is the interaction (G-by-E) between a particular segregant (genotype) and growth condition (environment). Since the available data was normalised against a control medium, there was (as expected) no significant contrition by *E*. The relative contributions to the total growth (phenotypic) variance from G and G-by-E were estimated by their respective sum of squares (Sum of Square for *id* is calculated as $\sum_1^p (id_i - \overline{id})^2$ and Sum of Square for the interaction *id*E* is calculated as $\sum_1^p (id_i * E_j - \overline{id_i * E_j})^2$)

### Defining a set of independently associated additive growth loci

A set of across-environment growth loci was defined. First, QTLs detected in the earlier environment-separate analyses with peak associations within 20kb and in pairwise $r^2 > 0.9$ were selected. Second, all the loci selected in step 1 were subjected to a multi-locus polygenic association analysis [50,51] to identify a final set of statistically independent loci (FDR < 0.05) with additive effects on growth in each tested environment [30]. Alternative definitions ranging from physical distance < 20 kb and $r^2 > 0.6$ to physical distance < 10 kb and $r^2 > 0.9$ were evaluated and found to result in very similar results in practice.

### Across environment evaluation of the additive growth loci

Several growth loci in the final set defined above only had significant individual associations in one growth environment. To test if they, as a group, contributed to the polygenic inheritance of growth also in other environments we compared the fit of the following models to the data (models 2 and 3) using a likelihood ratio test.

$$\overline{Y} = X_1\beta_1 + e \tag{2}$$

$$\overline{Y} = X_1\beta_1 + X_2\beta_2 + e \tag{3}$$

Here, $\overline{Y}$ is a vector of the average growth of each segregant (genotype) in a particular environment and $e$ is the normally distributed residual. The joint contributions by the individually significant/non-significant loci in a specific environment was modelled in $X_1\beta_1/X_2\beta_2$, respectively. $X_1$ includes a column vector of 1's for the population mean and column vectors with the genotype of each significantly associated SNP in the environment with the two homozygous genotypes coded as 0/2, respectively. $X_2$ includes column vectors with genotypes of all loci in the set defined above that was not individually significant in the tested environment. $\beta_1/\beta_2$ are vectors including the estimated additive effects for the two sets of loci. A likelihood ratio test was used to compare the fit of the two models using the *lrtest* function in R package *lmtest* [52].

## Detect individual loci involved in genotype by environment interactions

All growth loci defined in the polygenic analysis were evaluated for genotype by environment interactions. This by fitting the following two models to the data:

$$y_{ijk} = u + E_j + a_1b_1 + a_2b_2 + \cdots a_nb_n + e \tag{4}$$

$$y_{ijk} = u + E_j + a_1b_1 + a_2b_2 + \cdots a_nb_n + E_ja_xB_x + e \tag{5}$$

In both models, $y_{ijk}$ is the growth of replicate $k$ for segregant $i$ in environment $j$ ($j = 1..20$ environments; $k = 1..n_{ij}$; $n_{ij}$ is the number of replicates for individual $i$ in environment $j$); $a_x$ is the indicator regression variable for the genotype of QTL $x$ coded as 0 and 2 for the homozygous minor and major alleles; $b_x$ are the corresponding estimated additive effects; $u$ is the population mean and $E_j$ is the effect of environment $j$ (j = 1...20) on growth. Model 5 also includes an interaction term $E_ja_xB_x$ between one of the QTL and the environment. Model 5 was fitted for each QTL one at a time to test for its interaction with the 20 growth environments. The significance of each QTL by environment interaction was evaluated using a likelihood ratio test between models 4 and 5. Polygenicity was accounted for by the simultaneous fitting of all mapped loci in the two models. The analyses were performed using custom R scripts [53].

## Estimating the additive effects of QTL in different growth environments

A linear model (model 6) was used to estimate the additive effects of all the additive loci selected in the polygenic analysis in each tested environment.

$$\overline{Y} = X\beta + e \tag{6}$$

Here, $\overline{Y}$ is the average growth of the replicates for each segregant (genotype) in each tested condition. $e$ is the normally distributed residual. $X$ includes a column vector of 1's for the population mean and additional column vectors with the indicator regression variables for all the SNPs included in the model (coded as 0/2 for the two homozygous genotypes, respectively). $\beta$ is a column vector with the corresponding additive effects. This model was fitted for each environment independently to obtain estimates of the additive effects for each locus in each environment.

## Construction of an across-environment epistatic interaction network

A complete across-environment epistatic growth interaction network was constructed from the environment specific networks reported in Forsberg *et al* [31]. First, the environment specific networks inferred for each growth environment were extracted from the results of

*Forsberg et al* [31]. Second, we evaluated whether any of the pairwise interactions detected for a specific environment made significant contribution to growth at the remaining environments under a more lenient significant threshold only correcting for the total number of pairwise interactions detected across all environments. This was performed using a likelihood ratio test between models with and without the pairwise interaction for a particular pair of epistatic loci as described in detail by *Bloom et al* [30]. Then, the across-environment network was constructed by connecting the loci display pairwise interaction in any of the 20 environments using *igraph* [54] as descried in *Forsberg et al* [31]. These analyses were performed using custom scripts implemented in R [53] [will be made available upon publication or by request during review]. The raw data from which these were constructed are avaiable as supplement 5 in Bloom et al [30]; The analysis scripts/results from these earlier analyses are available at https://github.com/simfor/yeast-epistasis-paper and are described in detail by *Forsberg et al* [31]).

## Evaluation of the G-by-G-by-E interaction for a six-locus interaction network

The effects of a six locus interaction network, originally detected and explored for growth in IAA containing growth medium in Forsberg *et al* [31], were here evaluated across multiple environments. This analysis was performed to explore the association between the changes in activity of interactions of the epistatic genetic network and its contribution to growth in the different environments. To quantify the effects of G-by-G-by-E interactions, models 4 and 5 were fitted to the data with $a_x$ ($x = n = 1$) used as the indicator variable for each of the 64-genotype classes defined by the genotypes at the 6 loci. All other parameters, and the likelihood ratio test used to obtain the P-values for comparing the models, were the same as described above.

## Supporting information

**S1 Fig. Pairwise genetic correlation among growth measurements in the 20 used mediums.** Numbers in the cells are 100 times the genetic correlation, and environments were sorted based on their order after hierarchical clustering.
(TIF)

**S2 Fig. Joint epistatic network constructed by connecting shared loci from 20 epistatic networks detected for each media/environment.** Each dot in this plot is an epistatic QTL and the colour of the dot describes if the locus is detected with epistatic interaction for the current media with yes being blue or red (connected with more than 4 other loci) and no being grey. The pairwise interactions between loci are indicted by connected edges. The number of edges connecting two loci describe the number of times it is detected across 20 mediums, and the detected connection for current medium is highlighted with red (detected in Bloom et al,) and grey (other medias).
(TIF)

**S3 Fig. Pairwise overlap of loci detected with epistatic effects across the evaluated environments.** Numbers on the diagonal are the number of epistatic loci detected in a particular environment, and numbers in the cells are the number of overlapping epistatic loci between the pairs of environments. Phenotypes are sorted based on their order after hierarchical clustering.
(TIF)

**S4 Fig. Illustration of the phenotype resemblance and change of phenotypic correlation under different growth condition. A**). 3-dimentainal PCA plot of the yeast growth measured as the radial of colony on 20 different mediums. These mediums were made by adding small-chemical molelues to mimic different enviroments [30]. **B**). Pairwise Spearman rank correlation among growth measured on these 20 mediums. Numbers in the cell are 100 times the Spearman correation, and environments were sorted based on their order after hierarchical cluster.
(TIF)

**S5 Fig. Illustration of the connectivity of the 13 hubs.** 13 hubs connected with more than 4 loci in at least 1 environment is highlighted in red, loci epistatically interact with these hubs in at least one environment are labeled in yellow.
(TIF)

**S6 Fig. Changes in activity of interactions in the 13 epistatic networks across 20 environments.** In total, 13 epistatic networks were defined across 20 environments (represented in each column). Each row represents the activity of interactions in a particular network with corresponding hub alleles and candidate genes marked to the left. The colour intensity illustrate the proportion of loci, defined by their hub-QTL in a particular environment, that are connected as epistatic QTL (A) or additive QTL(B). The environments are sorted based on their order after hierarchical cluster.
(TIF)

**S7 Fig. Pairwise overlap of loci detected with additive effects.** Numbers on the diagonal are the numbers of additive loci detected for a particular enviroment, and numbers in the cell are the number of overlap addtive loci. Phenotype were sorted based on their order after hierarchical cluster.
(TIF)

**S8 Fig. Changes in genetic variances and broad/narrow sense heritabilities. A**). The phenotypic-, total genetic- and additive genetic variances for growth on 20 growth media. Total- and additive genetic variances were estimated as the product of the phentypic variance and the broad-/narrow-sense heritabilities, respectively, (panel **B**) from Bloom et al [30].
(TIF)

**S9 Fig. Releationship between the mean growth rank across 20 enviroments and the number of non-conpacitated alleles.** X-axis is the number of non-compacitated alleles across 13 hubs detected in our study, and y-axis is the mean growth rank obtained by first rank the growth measurements across 20 enviroemtns and then taking the artihmatical mean.
(TIF)

**S1 Table. Summary of the P values from a likelihood ratio test comparing a full model with all 311 growth QTL detected for all environments and a reduced model with QTL only detected for focal environments.**
(XLSX)

**S2 Table. Summary of the QTL by E analysis for 311 growth QTL.**
(XLSX)

**S1 Note. Evaluation of the independence of 130 epistatic loci.**
(PDF)

## Acknowledgments

We thank Simon Forsberg, Tilman Rönneburg and Thibaut Payen for discussions and comments on the manuscripts and figures.

## Author Contributions

**Conceptualization:** Yanjun Zan, Örjan Carlborg.

**Data curation:** Yanjun Zan.

**Formal analysis:** Yanjun Zan.

**Funding acquisition:** Örjan Carlborg.

**Investigation:** Yanjun Zan, Örjan Carlborg.

**Methodology:** Yanjun Zan, Örjan Carlborg.

**Project administration:** Örjan Carlborg.

**Resources:** Yanjun Zan, Örjan Carlborg.

**Software:** Yanjun Zan.

**Supervision:** Örjan Carlborg.

**Validation:** Yanjun Zan.

**Visualization:** Yanjun Zan.

**Writing – original draft:** Yanjun Zan, Örjan Carlborg.

**Writing – review & editing:** Yanjun Zan, Örjan Carlborg.

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
