## [Decision Letter · Decision Letter 0]

17 Feb 2020

Dear Dr zan,

Thank you very much for submitting your Research Article entitled 'Dynamic genetic architecture of yeast growth response to environmental perturbation shed light on origin of hidden genetic variation' to PLOS Genetics. Your manuscript was fully evaluated at the editorial level and by independent peer reviewers. The reviewers appreciated the attention to an important topic but identified some aspects of the manuscript that should be improved.

All three reviewers are highly qualified to evaluate the manuscript.  Nonetheless, whereas Reviewers #1 and #3 were quite positive, Reviewer #2 was significantly more skeptical.  Reviewer #2’s primary concern is that growth was measured by colony radius which is a phenotype that may have multiple confounding inputs.  This is a reasonable question.  As an aside, the editors noted that the tone of these comments was at times overly-aggressive, but that tone did not influence our assessment of the underlying scientific/technical issues. Can you use an independent measurement to validate the robustness of colony radius as a measure of growth in your experiments?  Barring that, I think it would be important to explicitly discuss this caveat when discussing the conclusions of the manuscript.

We therefore ask you to modify the manuscript according to the review recommendations before we can consider your manuscript for acceptance. Your revisions should address the specific points made by each reviewer.

[LINK]

Yours sincerely,

Gregory P. Copenhaver

Editor-in-Chief

PLOS Genetics

Reviewer's Responses to Questions

**Comments to the Authors:**

Reviewer #1: In this manuscript, the authors reanalyze a large existing dataset to work to better understand the potential role of epistasis in influence quantitative genetic variation in growth across a number of environments in yeast. This contributes to the growing level of data showing that GxGxE has an underappreciated contribution to quantitative traits. I only have a few suggestions.

In Figure 2E, do the authors have an estimate on how the additive variance explained responds to decreasing the number of loci? I ask because with 130 loci and the number of progeny utilized, the loci may have lost their independence and the fraction of additive variance may have asymptoted with a smaller collection of loci. Essentially, using 130 loci may be the equivalent of bar coding each individual and thus the additive estimate is functioning as a broad sense estimate at this level. Some discussion on this possibility would help the reader who is disinclined from epistasis to get past a high additive explanatory potential.

Is it that the environment is rewiring the network or altering how the output of the network is displayed? Rewiring implies (not inherently the authors intended implication) that the molecular connections are changing. I might suggest a different non-molecular wording.

Am I correct in interpreting Figure 5A as implying that the idea of rare alleles tending to have large effects needs to be tempered to say that they might also tend to be more environmentally and epistatically dependent? If that is correct, is there a reason to at best only provide veiled allusions to this outcome?

Editorial comments

As a personal thought, I tend to prefer cryptic genetic variance as crypsis can be a passive process while hidden implies effort/mechanism to hide. I know that the community has not settled on a nomenclature so I won’t attempt to enforce one.

Line 67, I believe the authors mean E. coli, S. cerevisiae and A. thaliana as at least one of the ensuing manuscripts is in Arabidopsis. And I do believe that the plant citation works to investigate some of the quant gen parameters discussed in line 91-94. Albeit in far narrower GxGxE collection.

If I’m reading correctly, all the multi-locus epistasis citations are in yeast. There are some papers on higher order epistasis in Plants and Drosophila that would help to move the conclusions beyond yeast.

In Line 420, I’m struggling with this equation Vtotal= H2 *Vp = H2. Is this correct as it seems odd to say broad sense heritability times Vp is equal to broad sense heritability. What am I missing?

Reviewer #2: This ms. aims to examine epistatic (GxG) and genotype-by-environment (GxE) interactions using a previously published data set (Bloom et al. 2015) consisting of large mapping population of segregants in the budding yeast S. cerevisiae whose growth was measured under a large number of environmental conditions and stressors.

The novel angle the authors employ, relative to two earlier studies using the same dataset (Bloom et al., Forsberg et al. 2017), is to treat growth as a single complex trait, rather than distinct traits for each environment, and use this single trait to compare "reorganization" of epistatic interactions across environments.

Unfortunately, while this is a clever idea in the abstract, the biological reality of endpoint growth measures (the trait measured in the original Bloom et al. study) makes this highly questionable in practice, and calls into question whether any useful biological insights are likely to emerge from such an analysis. Bloom and co-authors simply measured colony radii for each segregants after 48 hrs of growth on various agar plates. Colony radius reflects a mix of various cell biological and physiological traits -- division rate, cell size and cell shape, cell-cell interactions, etc. While various measures of growth on agar plates is often the microbiologists primary screening tool, no careful microbial geneticist would presume a priori that growth under neomycin stress is comparable to growth in xylose (to take as an example two of the more physiologically distinct conditions in the Bloom et al. data). While I don't doubt that some of the subsets of environments in the data affect underlying signaling and regulatory networks in similar ways, and hence would in some sense be comparable traits, this seems like a hypothesis to be tested not taken a priori to be true.

The difficulty of biological interpretability of the analyses is apparent from the discussion section -- little biological insight emerges about genetic networks in yeast, nor is there concrete new understanding the reader can take away about the genetic features that may shape GxGxE interactions in yeast or other organisms.

Reviewer #3: In this manuscript, the authors investigated and evaluated the role of G-by-G-by-E interaction on the complex traits. To explore the role of environment influencing epistatis that can lead to the expression of cryptic variation, the authors reanalyzed a dataset of 4,390 yeast genotyped segregants, which were phenotyped in 20 different conditions. As previously shown, they found that genetic interaction networks are contributing to complex traits. But more interesting, they highlighted and illustrated the importance of dynamics in epistatic network, genetic effects of individual alleles and genetic variance expressed across environments.

Overall, the manuscript is very interesting and bring a new insights into the mechanisms involved in the variation of complex traits.

I only have a couple of points that need to be addressed:

The authors use one cross between two given isolates, namely BY and RM. It is always difficult to know if the obtained results are not specific to the studied cross. This is something that should be discussed in the manuscript.

As the phenotypes are pre-normalized growth measurements (against growth in a control medium), the authors do not have a direct estimation of G-by-E. Did they try to run the same analysis on the non-normalized dataset? What are the results?

In the author summary, the word ‘gene’ is misused –

e.g. ‘individual with the same gene might have distinctive phenotypes’.

The discussion is very long and wordy. It could be shortened.

**Have all data underlying the figures and results presented in the manuscript been provided?**

Reviewer #1: Yes

Reviewer #2: Yes

Reviewer #3: Yes

PLOS authors have the option to publish the peer review history of their article (what does this mean?). If published, this will include your full peer review and any attached files.

Reviewer #1: No

Reviewer #2: No

Reviewer #3: No

---

## [Decision Letter · Decision Letter 1]

13 Apr 2020

Dear Dr zan,

Thank you very much for submitting your Research Article entitled 'Dynamic genetic architecture of yeast response to environmental perturbation shed light on origin of cryptic genetic variation' to PLOS Genetics. Your manuscript was fully evaluated at the editorial level and by two of the original three independent peer reviewers. The reviewers appreciated the attention to an important topic but identified some aspects of the manuscript that should be improved.

The reviewers are split in their opinion.  Reviewer #2 still thinks that treating growth under all conditions as a single phenotype misses critical underlying complexity and in doing so renders the analysis flawed.  Reviewer #1 is less concerned by this limitation and acknowledges that you aren't trying to explain all of what is happening but rather you're trying to use a single measurable trait to study complex genetics in a way that it not well represented in the literature. It is important to note that both reviewers are exceptionally qualified experts in the field, so I can appreciate both of their perspectives. As a way of threading the needle I think it will be critical to fully acknowledge the limitations described by Reviewer #2 in the text of the article and explain, at least by fully discussing if not by revising the analysis, how the system would be structured if individual growth components were accounted for. 

We therefore ask you to modify the manuscript according to the review recommendations before we can consider your manuscript for acceptance. Your revisions should address the specific points made by each reviewer.

[LINK]

Yours sincerely,

Gregory P. Copenhaver

Editor-in-Chief

PLOS Genetics

Reviewer's Responses to Questions

**Comments to the Authors:**

Reviewer #1: The authors have responded to and clarified my concerns from the previous version.

I have one suggestion, would it be possible to include the additivity and barcoding/independence analysis as a supplemental section. This analysis will be very helpful for the community when considering population size and locus independence when creating future populations or analyzing existing populations. I personally had not expected the role of environment in structuring the asymptote. This would be very useful for citations and guiding grant proposals.

Reviewer #2: Please see review in attachment.

**Have all data underlying the figures and results presented in the manuscript been provided?**

Reviewer #1: Yes

Reviewer #2: None

PLOS authors have the option to publish the peer review history of their article (what does this mean?). If published, this will include your full peer review and any attached files.

Reviewer #1: No

Reviewer #2: No

---

## [Editor Report · Decision Letter 2]

27 Apr 2020

Dear Dr zan,

We are pleased to inform you that your manuscript entitled "Dynamic genetic architecture of yeast response to environmental perturbation shed light on origin of cryptic genetic variation" has been editorially accepted for publication in PLOS Genetics. Congratulations!

Yours sincerely,

Gregory P. Copenhaver

Editor-in-Chief

PLOS Genetics

Comments from the reviewers (if applicable):

**Data Deposition**

http://datadryad.org/submit?journalID=pgenetics&manu=PGENETICS-D-19-02048R2

**Press Queries**

---

## [Editor Report · Acceptance letter]

1 May 2020

PGENETICS-D-19-02048R2 

Dynamic genetic architecture of yeast response to environmental perturbation shed light on origin of cryptic genetic variation 

Dear Dr Zan, 

We are pleased to inform you that your manuscript entitled "Dynamic genetic architecture of yeast response to environmental perturbation shed light on origin of cryptic genetic variation" has been formally accepted for publication in PLOS Genetics! Your manuscript is now with our production department and you will be notified of the publication date in due course.

With kind regards,

Matt Lyles

PLOS Genetics

On behalf of:
